# Diatoms Si Uptake Capacity Drives Carbon Export In Coastal Upwelling Systems

Abrantes F.[1,2]*, Cermeno P[3], Lopes C.[1,2], Romero O.[4], Matos, L.[1,4], Van Iperen J.[5], Rufino M.[1,2] and Magalhães V.[1]

[1]Portuguese Institute for the Ocean and Atmosphere, RuaAlferedoMagalhãesRamalho 6, 1495-006 Lisboa, Portugal. TM 351934309002, fatima.abrantes@ipma.pt

[2]Centro de Ciências do Mar (CCMAR--LA),Universidade do Algarve, Campus de Gambelas, 8005-139 Faro, Portugal

[3]Institute of Marine Sciences (ICM-CSIC),PasseioMarítim de la Barceloneta, 37-49. E-08003 Barcelona, Spain

[4]Center for Marine Environmental Sciences University of Bremen(MARUM), Leobener Str. D-28359 Bremen Germany

[5]Royal Netherlands Institute for Sea Research (NIOZ), Landsdiep 41797 SZ 't Horntje (Texel), The Netherlans

*Correspondence to*: F Abrantes (fatima.abrantes@ipma.pt)

## Abstract

Coastal upwelling systems account for approximately half of global ocean primary production and contribute disproportionately to biologically driven carbon sequestration. Diatoms, silica–precipitating microalgae, constitute the dominant phytoplankton in these productive regions, and their abundance and assemblage composition in the sedimentary record is considered one of the best proxies for primary production. The study of the sedimentary diatom abundance (SDA) and total organic carbon content (TOC) in the five most important coastal upwelling systems of the modern ocean (Iberia-Canary, Benguela, Peru-Humboldt, California and Somalia-Oman) reveals a global-scale positive relationship between diatom production and organic carbon burial. The analysis of SDA in conjunction with environmental variables of coastal upwelling systems such as upwelling strength, satellite-derived net primary production and surface water nutrient concentrations shows different relations between SDA and primary production on the regional scale. At the global-scale, SDA appears modulated by the capacity of diatoms to take up silicic acid, which ultimately sets an upper limit to global export production in these ocean regions.

## 1 Introduction

Coastal upwelling zones exist along the eastern boundary currents of the Atlantic and Pacific Oceans. In these areas, alongshore equatorward winds and the Coriolis effect force surface waters to diverge offshore, giving rise to the upwelling of nutrient-rich deep waters into the surface. The five most important coastal upwelling systems of the modern ocean are associated with: (1) the Canary Current off northwest Africa including its extension to the Iberian Peninsula (Iberia-Canary); (2) the Benguela current along southwest Africa; (3)the Peru-Humboldt current off western south America (SE Pacific); (4)the California Current off western north America (NE Pacific) and the Somalia-Oman monsoon derived upwelling off the Arabian Peninsula(Barber and Smith, 1984;Hill et al., 1998). At present, oceanic primary production represents 50% of Earth's primary production, and its highest contribution originates from the ±1% of the ocean characterized by coastal upwelling (Field et al., 1998; Hill et al., 1998).

The potential role of coastal upwelling systems on carbon uptake and export turns their dominant phytoplankton, the diatoms, into key players of marine planktonic food webs and the carbon cycle (Field et al., 1998). Diatoms require dissolved silica (as orthosilicic acid, $H_4SiO_4$), which they precipitate from seawater, to form their opal (amorphous hydrated Si) frustules (Lewin and Guillard, 1963) making coastal upwelling areas also a major Si sink (Tréguer and De La Rocha, 2013).

The prolific diatom-dominated blooms that respond to episodic inputs of nutrients are dominated by chain-forming marine species of the genus *Chaetoceros* Ehrenberg (Estrada and Blasco, 1985). Maximum fluxes of this genus coincide with intervals of increased nutrient availability in most coastal upwelling regions (California (Sancetta, 1995); Humboldt (Romero et al., 2001);Iberia-Canary (Wefer and G., 1993; Lange et al., 1994; Treppke et al., 1996b; Abrantes et al., 2002); Benguela (Treppke et al., 1996a; Romero et al., 2002); Somalia-Oman (Nair et al., 1989; Koning et al., 2001)attesting for their great capacity for C export.

Although only a minor percentage (1–4%) of the initial diatom population gets preserved in the sediments (e.g. Takahashi et al., 1989; Sancetta, 1992; Abrantes et al., 2002), the fact that *Chaetoceros* species produce heavily silicified spores with highly sinking rates, leads to a sedimentary assemblage dominated by the water-column blooming species, furthermore, the positive relationship between abundance of diatom frustules and content of organic carbon in sediments beneath major upwelling systems ((*Iberia-Canary system* (Abrantes, 1988; Nave et al., 2001)*; Benguela System* (Schuette and Schrader, 1981); *Peru-Humboldt system* (Abrantes, 2004)*, N California system*(Lopes et al., 2006))suggests that estimates of sedimentary diatom abundance (#valves/g - SDA) is representative of C export production at coastal upwelling regions. In support of this relationship, time-series of organic carbon content and diatom abundance from Atlantic coastal upwelling sites indicate concomitant variations of the two productivity proxies (e.g. Muller and Suess, 1979; Abrantes, 2000). Conversely, other sequences from the NW and SW African margins show contradictory results (e.g. Caulet et al., 1992; Heinze and Wefer, 1992; Martinez et al., 1999). A time-series located off the Oregon coast (Lopes et al., 2014), also shows disagreement between carbon and diatom-based estimates of primary productivity. In this record high abundances of small diatoms, which dominate highly productive ecosystems, co-occur with low concentrations of total organic carbon (TOC) in the sediments while higher carbon burial rates seem to be associated with the accumulation of large diatoms typical of unproductive environments.

It has long been recognized that both the physical and chemical processes as well as the biological response in coastal upwelling have a non-linear and complex relationship (Barber and Smith, 1981). Furthermore, intra- and inter-regional variability is known to be extensive through space and time, demanding dynamical and fine-scale biological modeling to faithfully understand those complex interactions. Here we analyze an extensive dataset of sedimentary diatom abundance (SDA for 703 sites) and organic carbon content (TOC for 200 sites) in concert with mean annual values of net primary production; upwelling index; and seawater nutrient concentrations extracted from long time-series and global climatologies

to assess the main factors controlling SDA. Our study relies on two main arguments: 1) it is possible to make generalizations about the functioning of coastal upwelling ecosystems (Barber and Smith, 1981); and 2) the sediments retain the imprint of the different processes that, on average, determine the physical, chemical and biological properties of coastal upwelling ecosystems.

## 2 Materials and Methods

The used data set comprises of 703 sediment samples recovered from the five most important coastal upwelling areas of the modern ocean, collected by different institutions and sampling processes (box-, multi- and in some cases piston- coring devices since the 1970's). For the analysis, the topmost sediment (0-1or-2 cm)was used (Figure1; Data available at

PANGEA). Calibration between the diatom data produced by different researchers followed the process described by Abrantes et al., (2005),and counting principles defined by Schrader and Gersonde (1978). Obtained values are given as number of valves / g of dry-sediment (#valves/g) and show 4 orders of magnitude in variability, between $10^5$ and $10^9$. Considering that cell growth is exponential, and to avoid the effect of extreme values and preserve minor variability within each order of magnitude, the data is plotted as its natural log (ln).

Organic Carbon content (TOC – w%) for the Galicia, Portuguese Margin, NW Africa – Canary, SE and NE Pacific was measured following the methodology in use at the IPMA sedimentology and micropalentology laboratory.Three replicates of 2 mg subsamples of each dried and homogenized sediment sample are measured, before and after combustion, on a CHNS-932 LECO elemental analyzer. The relative precision of repeated measurements of both samples and standards was 0.03 w%.

Upwelling indices were determined according to the National Oceanic and Atmospheric Administration (NOAA) procedure (Bakun, 1973) http://las.pfeg.noaa.gov/las/doc/global_upwell.html. Geostrophic winds were calculated from the FNMOC 6-hourly pressure analysis (1º grid) and further used to calculate wind stress and Ekman transport. Finally, the Ekman transport is rotated to get the offshore component (provided that the orientation of the coast is known), and a monthly upwelling index was obtained for each site location from 1969 to 2013.

Net Primary Production (NPP) data was provided by the Oregon State University Ocean Productivity Center (http://www.science.oregonstate.edu/ocean.productivity/) as mean annual and seasonal values for the available 10-year dataset (2003-2013). NPP is determined by the Vertically Generalized Production Model (VGPM) and is a function of chlorophyll, available light, and the photosynthetic efficiency (Behrenfeld and Falkowski, 1997).

The *World Ocean Atlas* 2013 (WOA13) is the source for *in situ* measured phosphate, silicate, and nitrate in µMat standard

depth levels and for annual, seasonal, and monthly composite periods(Garcia et al., 2014).Surface values correspond to the 10 m water depth and bottom data comprises the closest available value for each sample site water depth.

To address the long-term global relationship between SDA and all the environmental variables, escaping the temporal variability (seasonal and annual) within and between regions, we used the annual mean value estimated from the number of years of available data. Regressions between SDA and each variable annual mean were calculated and the corresponding

plot generated for the entire dataset as well as for the sites confined to the main upwelling zone of each system. The latitudinal boundaries for each main upwelling area were set to include seasonal upwelling regions and followed Rykaczewski et al., (2015).

## 3 Results And Discussion

### 3.1 Sedimentary Organic Carbon Content (TOC) and Diatom Abundance (SDA)

Our dataset allows evaluation of the relationship between SDA and TOC per geographic region and globally (200 sites; Table 1 and Figure 2). No relationship exists for the Canary upwelling System (Galicia to Canaries), the Benguela and the Peru-Humboldt systems, but a positive and significant relationship was found for the California system (NW Pacific) ($R^2$ =

0.98, $p$=0.01, n=5). When the entire existing dataset is considered, a significant positive relation is obtained ($R^2 = 0.40$, $p$=0.01, n=200). These results suggest that small-scale regional conditions are likely to determine different SDA and TOC accumulation loci. Diatoms give a detailed picture of the centers of maximum annual primary production generated during the upwelling season (Nelson et al., 1995; Abrantes and Moita, 1999). TOC is indicative of the mean annual production (independent of the generating process), and a stronger SDA-TOC relationship occurs in centers of preferential accumulation of fine-grained sediments, such as off large river mouths (e.g. Reimers and Suess, 1983; Abrantes, 1988; Abrantes et al., 2004). On a global scale, the covariance between SDA and TOC is likely to reflect the higher annual contribution of diatoms to total carbon export in the high-Si coastal upwelling systems.

When dealing with sediment components, however, one has to verify to what extent can sedimentary processes alter the sediment record. Both diatom and organic carbon burial success depend on, (1) the sealing effect provided by the rate of contribution of other particles to the bulk sediment, that is, sediment accumulation rate (MAR); and, (2) early diagenetic processes on the seafloor (Broecker and Peng, 1982).

### 3.1.1 Sealing effect

MAR corresponds to the amount of sediment deposited by unit of area and time, is expressed in g cm$^{-2}$ ky$^{-1}$, can only be calculated if knowing the age of the sediments and it is subject to errors that can be caused by faulty age models, or processes like sediment focusing by bottom currents. Such problems can only be circumvented through the combined use of MAR estimation from excess $^{230}$Th determinations (Lyle et al., 2014). For our sites, little or no $^{230}$Th data is available, and dated cores are scarce and punctual, reducing the original data set to 28 sites (SI Table 2). We used this data to investigate the possible influence of MAR variability on diatom accumulation rate (DAR)according to:

DAR (# valves cm$^{-2}$ ky$^{-1}$) = Diatom abundance (SDA - #valves/g) * MAR (g cm$^{-2}$ ky$^{-1}$)

where MAR (g cm$^{-2}$ ky$^{-1}$) is calculated multiplying the sedimentation rate (SR, cm/yr) by the sediment dry bulk density (g/cm$^3$).

Sediment dry bulk density is also not available for the majority of the dated cores, but one can consider its variability between 0.7 and 2 g/cm$^3$ (authors laboratory data for thousands of samples). SR was determined on the basis of the two topmost published radiocarbon dates for each site (SI Table 2). Both DAR and MAR were estimated using the minimum, mean and maximum dry bulk density (0.7, 1.4 and 2 g/cm$^3$ -SI Figure 1A, B). The results reveal that SDA is not dependent on MAR ($R^2$=0.01), but DAR values are mainly controlled by diatom abundance ($R^2$=0.90).

When the same exercise is performed for TOC concentrations, the data set gets reduced to 10 sites which is not representative of all 5 areas and displays no relation to either MAR ($R^2$=0.18 at $p$=0.1 and n=10),or to TOC accumulation rate ($R^2$=0.62 at $p$=0.1 and n=10).

### 3.1.2 Diatom and TOC diagenetic effects at the sea floor

Diatom dissolution on the seafloor, although influenced by factors such as pH, temperature, organic matter and metal ions associated with the frustule surface, is mainly determined by the degree of Si saturation in the medium, that is, the concentration of silicic acid in the bottom waters (e.g. Lewin and Guillard, 1963;Hurd and Takahashi, 1982; Van Bennekom and Berger, 1984; Van Cappellen et al., 2002).

No relation between SDA and concentration of silicic acid in deep waters was found at the regional level (SI Figure 2). Throughout the world's oceans, the pathway of thermohaline deep water circulation generates inter-basin differences in the concentration of silicic acid, which increases as deep waters flow from the Atlantic Ocean to the Southern and Pacific basins (Rageneau et al., 2000). The comparison of bottom [Si(OH)$_4$]to SDA at our sites displays the two expected clusters that separate the lower silicate of the Atlantic *vs.* the higher silicate of the Pacific (Figure3). Furthermore, the mean SDA (ln transformed) values for the two clusters are statistically different (t test: T= -20.481, $p$<0.001, mean Atlantic=5.606 +- 0.072

and mean Pacific/Indian = 6.696 +- 0.076), but a large overlap within the diatom abundance of the two clusters is also evident. These results indicate that despite the diatom losses related to sea-floor $[Si(OH)_4]$, the preserved sediment diatom signal carries generalized information that overrides the result of dissolution effects.

As to TOC, the preservation of the organic material that escapes water-column degradation and reaches the bottom, is dependent on the type of organic matter, bioturbation level and bottom water oxygenation (Emerson, 1985; Canfield, 1994; Zonneveld et al., 2010). Aside from the long-standing debate regarding the kinetics of organic matter decomposition at different bottom-water $O_2$concentration, Canfield (1994) showed that in regions with high rates of organic carbon deposition (typical of continental margins), differences in preservation are weakly dependent on bottom water $O_2$ concentrations.

## 3.2 Upwelling Intensity and SDA

In the modern subtropical ocean, coastal upwelling is mostly seasonal (Spring-Summer) but it can also be perennial at lower latitudes such as off Cape Blanc (Canary system), Walvis-Bay (Benguela system), off Peru (Humboldt system). Inter-annual atmospheric and oceanic variations can cause latitudinal extension and retraction in the seasonal spatial coverage of each system as well as in the offshore extension of upwelling filaments. The best-known and dramatic inter-annual variations in coastal upwelling derived primary production occurs in the Humboldt-Peru system, caused by the El Niño-Southern Oscillation (ENSO) Cycle (Barber and Kogelschatz, 1990; Philander, 1990).

To investigate the possible effect of upwelling strength on sediment diatom abundance, we used the 47-year time-series of the annually averaged upwelling index estimated for each sample site following the method of Bakun (1973). A method that can generate errors for sites located close to the equator or within 1 degree off the coast, that is, the inner-shore areas which are the most influenced by the coastal upwelling process, and where a large number of our sites are located (519 in total – SI Figure 3). An alternative approach would be the use of local wind datasets, but problems arise from the fact that different areas have dissimilar data sets both in terms of time length as in the index calculation approach. As such, we decided to use the NOAA dataset after exclusion of very large values. The results (SI Figure4; Table 1) reveal that independently of using all the sites or just the main upwelling areas, upwelling index is only significantly related to SDA in the Southern Hemispheric systems (Benguela and Humboldt – Table 1). As a whole and at the scale represented by the sediments (tens – hundreds yr), the physics (although essential), does not appear as the primary factor determining diatom bloom size and its sediment record.

## 3.3 Primary production and SDA

On the global scale, the restricted coastal upwelling areas are always highly associated to the spatial distribution of primary production, independently of the methodological approach (Berger, 1989; Field et al., 1998; Gregg and Conkright, 2008). There is also evidence for a good agreement between net primary production (NPP) and sediment diatom abundance, as well as with the abundance of resting spores of *Chaetoceros*, the dominant genus in the sediment assemblage at the regional level (e.g. Figures 1E and 8 in Abrantes et al., 2005 and Figure 6 in Lopes et al., 2005).The relationship between the sediment diatom abundance data and NPP at each of this study sites is presented in figure 4A. Although the general positive relationship between SDA and NPP, a significant correlation is only observed for the Canary system region with perennial upwelling conditions (Table 1). Nonetheless, a close inspection of figure 4A reveals that potentially three different correlation lines can be defined through data from all different geographic locations. A result that reflects a combination of intra-regional differences and inter-regional similarities in the SDA-NPP relationship, but hampers the possibility of defining a single equation at the global level.

## 3.4 Nutrient Availability and SDA

The variables evaluated so far revealed that SDA, contrary to what has been found at the regional level, is not solely

influenced by upwelling strength or ecosystem productivity. Sedimentary processes do not also appear to be a major determinant of SDA. Thus, we explored the role of nutrient availability in regulating SDA under the premise that nutrient supply ratios can select for specific phytoplankton groups (Raymont, 1980; Barber and Kogelschatz, 1990; Platt et al., 1992). In coastal upwelling areas the type and amount of nutrients delivered to the ocean surface depends on the upwelling intensity (discussed above) since it determines the water mass source depth. Upwelled waters are in general, central waters of sub-polar and sub-tropical origin in different mixing proportions. On a global scale, the previously mentioned inter-ocean fractionation also causes different nutrient concentrations to exist in the Atlantic and the Pacific coastal upwelling source waters. Similarly, intra-ocean circulation determines nutrient differences at the hemispheric level, in particular for silicate concentrations (Broecker and Peng, 1982; Sarmiento et al., 2003).

In coastal upwelling areas, diatoms dominate the biomass of phytoplankton communities. Their obligatory requirement for silicic acid makes it a limiting nutrient for diatom growth (Brzezinski and Nelson, 1996;Lima et al., 2014). Mesocosm experiments revealed that marine diatoms dominate phytoplankton communities when the concentration of silicic acid exceeds 2µM whenever other nutrients are in excess (Egge and Aksnes, 1992). Further analyses have shown that diatom productivity is determined by the availability of silicic acid in broad regions of the global ocean (Dugdale and Wilkerson, 1998).

Besides macronutrients, micronutrients such as iron (Fe) are also essential to sustain elevated rates of primary production, not only in the high-nutrient low-chlorophyll ocean regions (HNLC) e.g. (Martin et al., 1990) but also in coastal upwelling systems (Hutchins et al., 1998); (Bruland et al., 2001); (Hutchins et al., 2002). Although the various potential Fe sources existing in all coastal upwelling systems; remineralization within the water column, episodic inputs of Fe from continental regions (riverine or eolian input) (Boyd and Ellwood, 2010), re-suspension from the benthic boundary layer and early diagenesis remineralization of sediment Fe and upward diffusion of Fe-enriched pore waters (Johnson et al., 1999); (Masay et al., 2014)), iron limitation is currently considered an important control on phytoplankton growth in the Californian and Peruvian upwelling zones, while the Canary and the Benguela systems are relatively iron-replete (Capote and Hutchins, 2013).

Fe limitation has been found to decrease diatom importance within the phytoplankton community (Hutchins, 1998);(DiTullio et al., 2005), limiting the extent of  blooms to large sized (> 40µM) diatoms such as *Coscinodiscus* (Bruland et al, 2001), and lead to thicker frustules (Pichevin et al., 2014), and  higher C export efficiency (Brzezinski et al., 2015). However, Fe enrichment experiments in the Pacific upwelling systems produced different results for different areas.(Franck et al., 2005) revealed a relative decrease in Si production for the Humboldt system and concluded that these regions were at first not Fe limited. DiTullio et al., (2005) consider that in upwelling systems, the luxurious Fe uptake by diatoms, near the coast, allow them to remain Fe replete while inhabiting low-Fe waters. Conversely, (Brzezinski et al., 2015) defend that upwelling brings inadequate iron to the surface for phytoplankton to completely utilize macronutrients and consider that although the Fe and Si co-limitation of diatom growth observed in the HNLC equatorial Pacific system (Si limiting diatom Si uptake rates and diatom silicification, and Fe limits diatom growth rate)(Brzezinski et al, 2008), is lower under strong coastal upwelling, it increases along aging waters. Furthermore, in the California Current system low Fe amplifies total net opal production in different ways depending on the Si:N ratio of the initial upwelled waters (Brzezinski et al., 2015). Such different results reflect the transient state of coastal upwelling systems as well as the uneven Fe distribution within and between regions (Hatta et al, 2014). Furthermore, growth rate of small diatoms was not altered even in the highly Fe limited Eastern Equatorial Pacific (Brzezinski et al., 2011), and coastal upwelling diatom communities are dominated by small (3-20 µM) *Chaetoceros* species known to produce heavily silicified resting spores under nutrient stress conditions.

The lack of well-distributed Fe data prevents us from assessing this micronutrient effect on SDA in the same manner as for other variables. Yet, spatial Fe variability is also accompanied by time variability (Pichevin, 2014) and as for all other variables, the sediment record reflects the dominant state throughout the time it represents (10 to 100 yr).

To evaluate the large-scale spatial and temporal relation between surface water macronutrient contents and SDAwe used mean annual $[NO_3^-]$, $[PO_4^{2-}]$ and $[Si(OH)_4]$ from theWOA13 database (Table 1 and SI figure5 and figure 4B).SDA shows a significant positive relationships to$[NO_3^-]$ and $[PO_4^{2-}]$on the Humboldt (0.44 and 0.52 respectively, at $p$=0.1 and n=162) and the Somalia-Oman systems (0.79 and 0.67 at $p$=0.1 and n=23), but not at the global level, as it is clear from SI figure 5.

When the data set is restricted to the main upwelling areas the positive relationship between SDA and nitrate becomes positive and significant (0.39 at $p$=0.1 and n=400), consistent with the effect of nitrate on diatom standing stocks. Between SDA and $[Si(OH)_4]$ we found a statistically significant relationship for the Canary, Humboldt and Somalia-Oman systems (0.52, 0.49 and 0.74 respectively –see Table 1). The relationship was also significant for the Benguela system (0.41 at $p$=0.1 and n=32)when only the main upwelling areas are considered (Table 1; Figure 4B).

Figure 4B and SI figure 6A represent the relationship found between SDA and silicate for the main upwelling sites and the total dataset respectively, and show the same pattern but a reduction in data scattering when only the main upwelling areas are considered (figure 4B). In detail, it revealsa steep slope near the low silicate concentrations ($\leq 2$ µM) defined by the NE Atlantic regions (Galicia to the Canary Islands region – 29ºN). An increase towards the asymptote is defined by the data from the Canary (Cape Blanc), Benguela and Humboldt systems, andthe sites from the Somalia-Oman and California

regions, the Si-richest within the modern coastal upwelling systems, establish the maximum diatom abundances. This pattern is retained when treating the Atlantic and Pacific Ocean datasets independently (SI Fig. 6B, C), which confirmsthat the inter-hemispheric difference in preservation potential is not the maincause behindthe globalSDA pattern.

These results led us to explore the possible link between SDA and $[Si(OH)_4]$ in the surface watersusing a theoretical framework focused on cell physiology. Our strategy assumes that the observed SDA reflects surface ocean diatom

productivity and export fluxes(Lisitzin, 1971); Tréguer and De La Rocha, 2013). The concentration of silicic acid in upwelled waters can potentially increase SDA in two ways:1) increasing diatom productivity (and hence diatom numerical abundance) and, 2) increasing the thickness and sinking rate of diatom frustules.

According to (Brzezinski et al., 2015) the ambient Si:N ratio determines in which way Fe influences the total net opal production, with higher ballast (diatom species and valves more silicified and resistant to dissolution) for low Si:N and

higher Si production (diatom valves less silicified and resistant to dissolution) for higher Si:N conditions. Following this logic we have investigated the influence of Si:N on DAR. No relationship was found, either regionally or for all the coastal upwelling systems. However, by averaging the SDA and annual Si:N values for the main upwelling sites in each coastal upwelling system, a highly significant and positive relationship emerges ($R^2$=0.64 for a n=338 and $p$=0.1). The SDA *versus* Si:N relationship throughout the 5 most important coastal upwelling areas (Fig. 5) follows a trend that is similar to that

observed between SDA and $[Si(OH)_4]$. Such results imply that on the long term (tens to hundreds of years), independently of [Fe] and its potential effect, it is the physiological response of marine diatoms to surface waters silicic acid availability thatcontrolscoastal upwelling SDA. It has been shown that not only does diatom productivity increase with increasing silicic acid concentrations, but also that diatoms have a higher $[Si(OH)_4]$ uptake rate for higher silicic acid ambient concentrations (Dugdale and Wilkerson, 1998); (Goering et al., 1973); (Nelson et al., 1981); (Dugdale et al., 2011). Thus, we speculate that

it is silicic acid concentration that sets the upper limit to diatom growth rate, which, in turn, is imposed by the physiological capacity of individual cells to take up and use silicic acid.

The dynamics of silicic acid uptake by diatoms has been a matter of continuous debate (Dugdale, 1967); (Goering et al., 1973); (Nelson et al., 1981); (Thamatrakoln and Hildebrand, 2008), but it is traditionally parameterized by a Michaelis-Menten model, defined by two parameters, the maximum uptake rate and the half saturation constant (i.e. the concentration

of nutrient at which the population reaches half of the maximum uptake rate), (Del Amo and Brzezinski, 1999). We applied the mathematical expression used to model nutrient uptake kinetics to our SDA data at the regional, oceanic, hemispheric and global scale:

$V = V_{max}([Si(OH)_{4max}] / (K_s + [Si(OH)_{4max}]))$

where $V_{MAX}$ is considered to be the maximum diatom abundance value ($Diat_{MAX}$) defined by the best line fit (log) to the data; [S] the concentration of silicic acid ($Si_{surfMAX}$) and Ks the half saturation constant defined as ($Diat_S$) = ($Diat_{MAX}/2$) (e.g. Fig.6). The results obtained are presented in SI table 1, where the parameter V represents the expected sediment diatom abundance at a given silicic acid concentration in the surface ocean waters. V values are in the order of $10^8$ for a silicic acid concentration ± 7µM $l^{-1}$ in the Pacific and Indian upwelling areas, as well as for all 5 areas together. For the Southern Hemisphere coastal upwelling systems the values were in the order of $10^7$ for the same $[Si(OH)_4]$, ± 7µM $l^{-1}$. The parameter V in the Atlantic Ocean is one order of magnitude lower than in the Southern Hemisphere systems and two orders of magnitude lower than in the Pacific Ocean or all the 5 areas together. Similarly, a $[Si(OH)_4]$ yield-dose of 1µM $l^{-1}$ in the Atlantic and North Hemisphere (excluding the Somalia-Oman system), suggests that in these oceanic regions marine diatoms are outcompeted by non-siliceous phytoplankton in agreement with observations in mesocosm experiments (Egge and Aksnes, 1992).

## 4 Conclusions

In summary, our results indicate that SDA is not dependent on sedimentation rate nor is it solely a function of the bottom water silicate concentration. On decadal to centennial time scales, nitrate and silicic acid concentrations in surface waters contribute to define the global SDA pattern. However, the ability of marine diatoms to take up silicic acid sets an upper limit to SDA that strongly influences the potential of coastal upwelling areas for C sequestration.

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

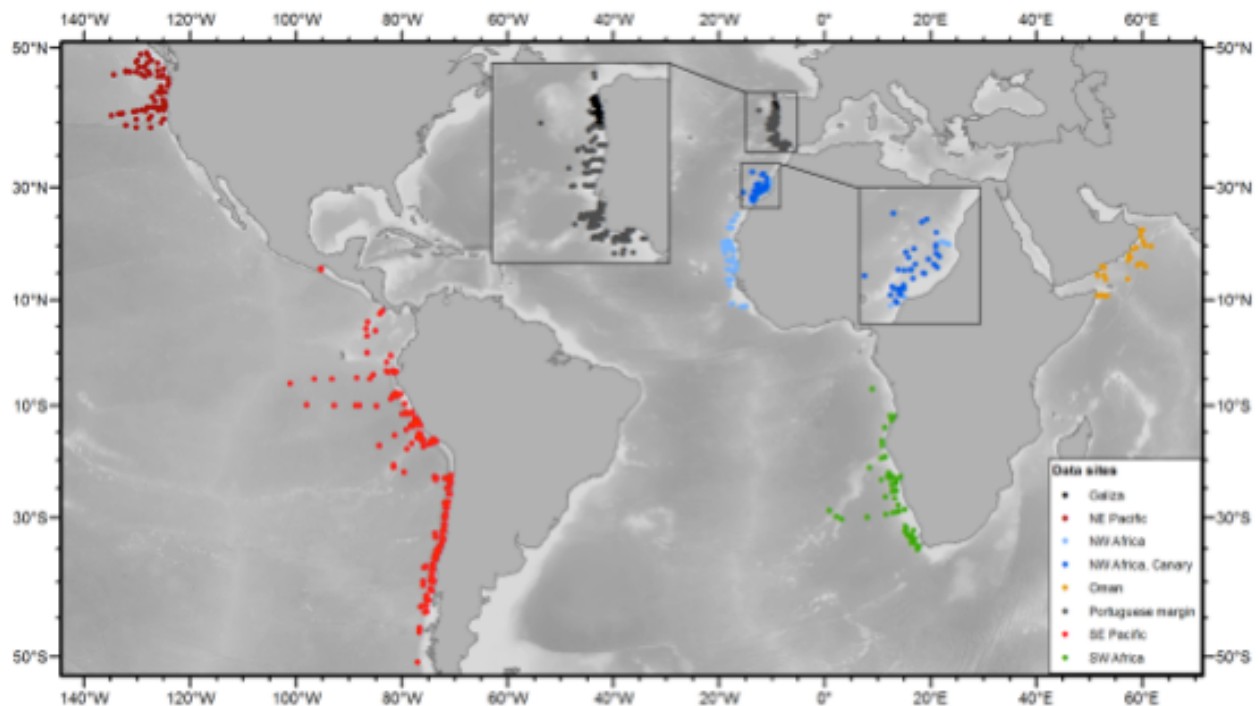

**Figure 1: Geographic distribution of the total data set sites considered in this study. Map in WGS84 and Mercator projection. Different colors represented different upwelling systems.**

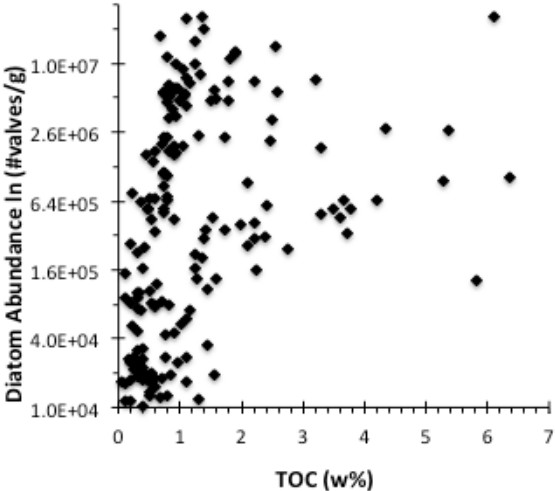

**Figure 2: Relationship between total marine diatom abundance ln (# valves /g) and total organic carbon content (W%) for the Canary (Galicia to Canary), Humboldt (SE Pacific) and California systems (NE Pacific).**

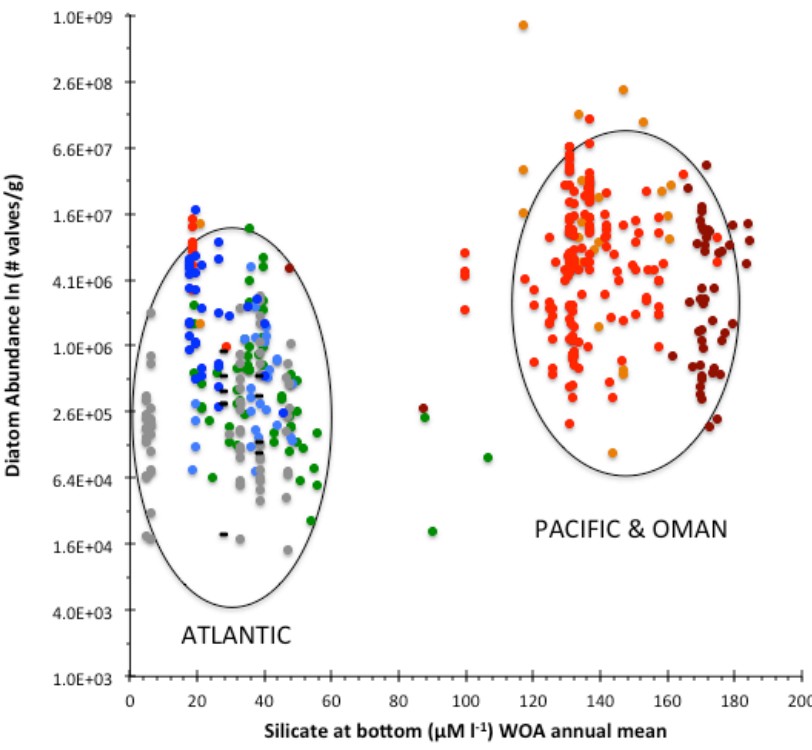

**Figure 3: Relationship between total marine diatom abundance ln (# valves /g) and bottom water [Si(OH)$_4^-$] in µM from WOA 2009 database. Color scheme as in Figure 1.**

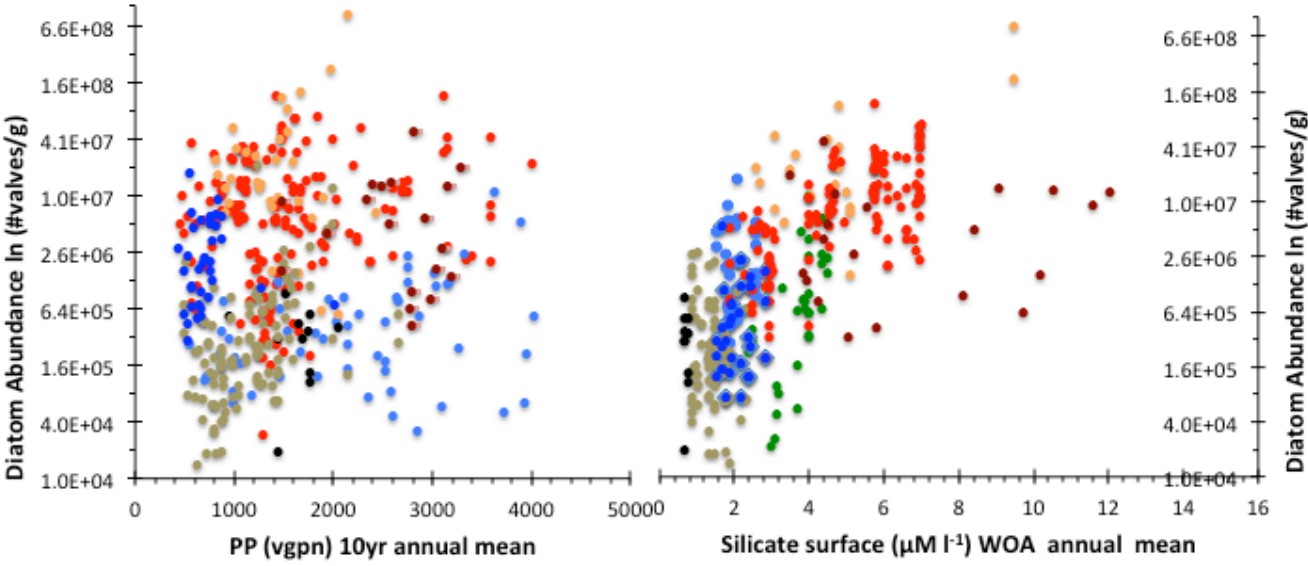

**Figure 4: Relationship between total marine diatom abundance ln (# valves /g) to: A. NPP generated from SeaWiFS chlorophyll distributions according to the Vertically Generalized Production Model (VGPM) (Behrenfeld and Falkowski, 1997); B. surface water [Si(OH)$_4^-$] in µM, from WOA 2013 database. Sites located on main upwelling areas. Color scheme as in Figure 1.**

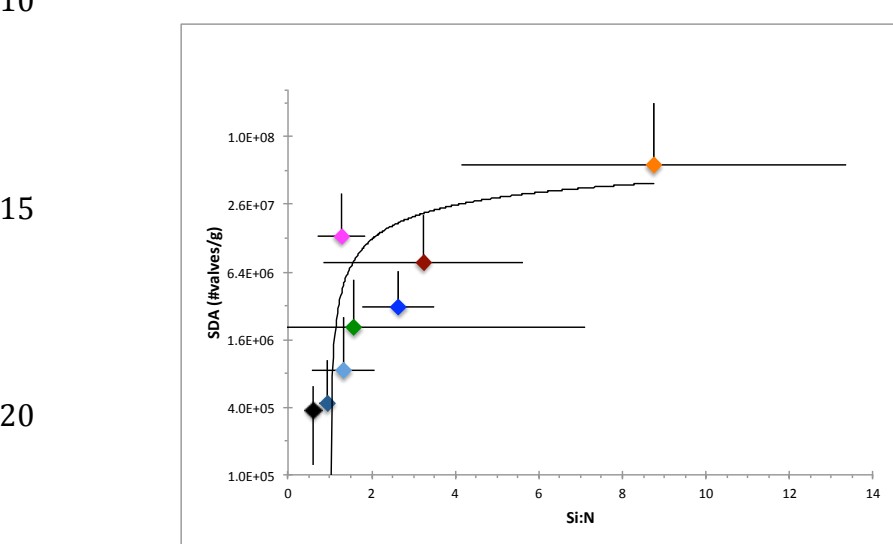

**Figure 5 – Relationship between average SDA ln (# valves /g) to average Si:N estimated from the WOA13 database per upwelling system and considering only the true upwelling sites. Bars represent StDev.  Color scheme as in Figure 1.**

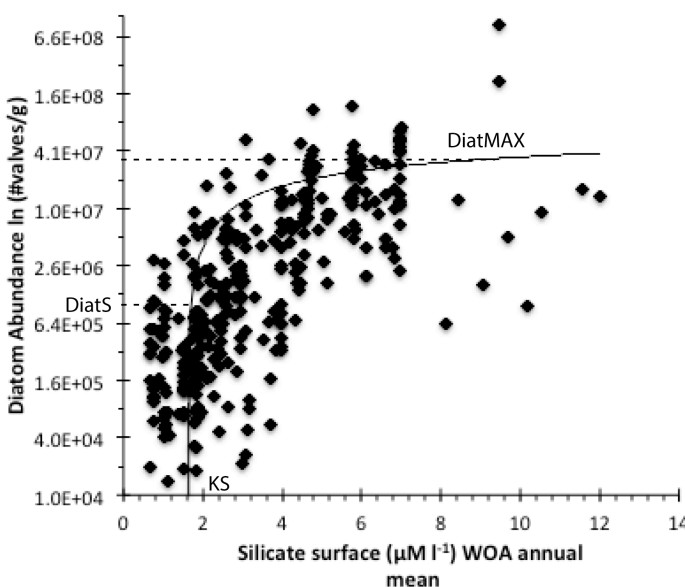

**Figure 6: Translation of Michaelis-Menten model, which defines the relationship between population abundance and [Nutrient] to**
10 **the long-term SDA dataset. That relationship is expressed by the mathematical equation $V = V_{max} [N] / (K_s + [N])$ where (V) is the**
**algal population growth rate, ($V_{max}$) the maximum diatom abundance in the sediments ($Diat_{MAX}$) at threshold $[N = Si_{surfMAX}]$ above**
**which population abundance remains constant; $[N - Si_{surfS}]$ is silicate concentration on upwelled surface waters, and ($K_s$) is silicate**
**concentration at $Diat_{MAX}/2$**

