# Peer review of "Diatoms Si Uptake Capacity Drives Carbon Export In Coastal Upwelling Systems"

_Biogeosciences, 2015_

## Referee Comment (RC1) · Anonymous Referee #2 · 18 Mar 2016

Under my point of view this work is supported by an important data set: 703 diatom samples, and also 200 sediment samples where TOC was analyzed. Authors also compare their data with other variables: Net primary production, phosphate, nitrate and silicate. This paper could be used an important tool for those researchers dealing with paleo-studies in diatom preservation versus productivity. Since authors deal with different concepts such as the sealing effect, the diatom and TOC diagenesis at bottom, upwelling intensity, primary production or nutrient availability vs diatom accumulation. the few technical corrections: Page 4, line 35: Figure 2 relates SCD/TOC, but author releted it with silicid acid table 1: (review Galiza)

---

## Referee Comment (RC2) · Anonymous Referee #1 · 21 Mar 2016

General Comments: This study synthesizes is a significant number of total organic carbon and sedimentary diatom abundance (SDA) samples from surface sediments in five important coastal upwelling regions around the globe. The authors attempt to use these data to determine whether a global generalization regarding the most important factor regulating SDA. This is a creative synthesis of data which is not only spatially and temporally expansive, but time consuming to produce (e.g. quantification of diatom valves in sediments).

Specific Comments: In its present form, I have concerns; but if addressed, could see this being an interesting contribution which supports the recent synthesis by Tréguer and De La Rocha (2013). In that synthesis, Tréguer and De La Rocha (2013) observed that the burial of diatoms is mainly driven by the magnitude of water-column biogenic silica production, due to the fate of nearly all produced diatom silica is dissolution in the

water column or sediments. Tréguer and De La Rocha (2013)'s data is based on silicon biogeochemical measurements; however, this study gets to a similar result using just sediment diatom valve abundance and annual mean nutrient data (i.e. more production, more SDA). The overall presentation is confusing in some areas and will need some refinement (e.g. full of undefined abbreviations, lack of calculation transparency and detail, important details in the supplementary material).

The idea of SDA in the sediments also assumes that these diatoms have a significant amount of carbon to drive at least a portion of TOC trends; however, iron limitation (see below) of diatom production produces cells with high Si and low organic matter diatoms. It has been known for 15 years that iron and macronutrients play a significant role in regulating diatom production in upwelling regions (e.g. Bruland et al. 2001, Limnology and Oceanography 46(7)). And because diatoms Si per cell is plastic, driven by both the rate of uptake (substrate dependent) and the duration of a cell cycle (growth-rate dependent), iron limitation can have a substantial effect on ballasting diatoms (i.e. high silicic acid means diatoms take up Si at high rates, low Fe means diatoms grow slower and decouple Si and C or N quotas). While the records for dissolved iron in the water column are not available, for thoroughness, it would be good to at least discuss iron as it affects both the Si quota for a diatom and also reduces the organic matter per cell.

Technical Corrections: General: the author order on citations is odd (e.g. sometimes by year, other times listed alphabetically by first author, other times no obvious pattern, Page 5, line 27). Page 2, Lines 9 – 10: Field et al. (1998) showed that highly productive regions (e.g. Coastal upwelling zones with satellite Chlorophyll a > 1 mg/m3) was only ∼18% of total ocean net primary production, not 80-90% as stated here. Page 3, Line 26: change $\mu$ML-1 to $\mu$M Section 3.1.2: the main points of this section could be clearer Page 4, Line 26: perform vs. preform Page 4, Line 42: please change [Si(OH)4-] to [Si(OH)4] (at seawater pH Si is not ionic), please make this correction elsewhere in the manuscript Page 4, Line 44: this sentence is basically what Tréguer and De La Rocha (2013) have shown among all oceanic systems. Considering so

much dissolution occurs in the water column and sediments (e.g. >95%), the burial of diatoms in sediments is largely predicted by the magnitude of their production in the surface waters. See General Comments. Page 5, Line 23-24: While this is certainly the conclusion of this analysis, such a statement is rather grandiose given the record is a 47-year average, and later it is discussed that the temporal scale of inference for the Silicon-related discussion is tens to hundreds of years (Page 6, line 32). Perhaps be more conservative? Page 5, Line 37: contrarily vs. contrary Page 6, Line 7: silicic acid "can be" a limiting nutrient for diatom growth, most kinetic data demonstrate it is unlikely for growth to be limited by silicic acid (also Brzezinski and Nelson 1996 reference was in the North Atlantic gyre, not an upwelling system, see below for upwelling systems references). Page 6, Line 9: the Dugdale and Wilkerson (1998) model was driven by silicic acid nutrient profiles, not direct data, plus this was in the equatorial Pacific not a coastal upwelling system. Additionally, in this same system, Brzezinski et al. (2008, Limnology & Oceanography) showed it was unlikely that silicic acid was limiting diatom growth based on the degree of kinetic limitation observed. Page 6, Line 29: This assumption may be reasonable (see comment on Page 4, Line 44 regarding Tréguer and De La Rocha 2013 study and perhaps cite this as justification) but completely ignores diatom frustules in the sediments which have been authigenically transformed via reverse weathering (e.g. Michalopoulos et al. 2000, Geology 28) and are likely to not be quantified (i.e. not recognizable). Additionally, the SDA proxy may be more representative of large and/or heavily silicified diatoms only. Perhaps the potential bias could be discussed in the methods. Page 6, Line 34: Dugdale et al. (2011) reference is from equatorial Pacific, not a coastal upwelling system (e.g. see Goering et al. 1973 DSR for Peru or Nelson et al. 1981 Consumption and Regeneration of Silicic Acid in Three Coastal Upwelling Systems for Baja California and Northwest Africa). Page 6, Line 37: Goering et al. 1973 actually showed Michaelis-Menten uptake fit Si uptake responses in an upwelling system, the Dugdale paper did not focus on silicate. Page 6, Line 42: expand [S] to be [$Si(OH)_4$] Page 7, Line 2: use of SisurfMAX is unclear until you mine through supplementary tables Page 7, line 18: I disagree, if iron is

the limiting nutrient in the upwelling system (e.g. Bruland et al. 2001, Limnology and Oceanography) then it isn't Silicon uptake which affects C sequestration, it is iron which leads to excess Si ballast (see also Brzezinski et al. 2015, JGR Oceans).
* * *

---

## Author Comment (AC1) · 29 Apr 2016

The authors thank the reviewer for her/his comments

Reply to Referee 2 comments "Page 4, line 35: Figure 2 relates SCD/TOC, but author related it with silicic acid table 1: (review Galiza)"

- The Spanish designation Galiza has been changed to the English one, Galicia

- The Figure 2 mentioned on Page 4, line 35 is the Supplementary Information Figure 2, rather than the Paper figure that the reviewer refers to.
* * *

---

## Author Comment (AC2) · 29 Apr 2016

The authors thank referee 1 by is careful reading of the discussion paper and thoughtful comments. Below we respond to the comments one by one.

The idea of SDA in the sediments also assumes that these diatoms have a significant amount of carbon to drive at least a portion of TOC trends; however, iron limitation (see below) of diatom production produces cells with high Si and low organic matter diatoms. It has been known for 15 years that iron and macronutrients play a significant role in regulating diatom production in upwelling regions (e.g. Bruland et al. 2001, Limnology and Oceanography 46(7)). And because diatoms Si per cell is plastic, driven by both the rate of uptake (substrate dependent) and the duration of a cell cycle (growth-rate dependent), iron limitation can have a substantial effect on ballasting diatoms (i.e. high

[Figure]

silicic acid means diatoms take up Si at high rates, low Fe means diatoms grow slower and decouple Si and C or N quotas). While the records for dissolved iron in the water column are not available, for thoroughness, it would be good to at least discuss iron as it affects both the Si quota for a diatom and also reduces the organic matter per cell.

- A discussion concerning the role of Fe is now included in the text. Furthermore, some other considerations are also presented at the end of this letter.

Technical Corrections: General: the author order on citations is odd (e.g. sometimes by year, other times listed alphabetically by first author, other times no obvious pattern, Page 5, line 27).

- This Issue has been verified and corrected. In text references are now chronologically listed;

Page 2, Lines 9 – 10: Field et al. (1998) showed that highly productive regions (e.g. Coastal upwelling zones with satellite Chlorophyll a > 1 mg/m3) was only 18% of total ocean net primary production, not 80-90% as stated here.

- This sentence reflects the findings of Field et al, (1998) that the Oceans contribute 46.2% of the global annual NPP and Hill et al, (1998), who considers that 80-90% of oceanic production is concentrated in the 1% area occupied by the highly productive eastern boundary currents' related coastal upwelling systems as a whole (>500 mg C m-2 yr-1), not just the very coastal areas. The sentence has been rewritten.

Page 3, Line 26: change _ML-1 to _M

- Corrected

Section 3.1.2: the main points of this section could be clearer

Page 4, Line 26: perform vs. preform

- Corrected

Page 4, Line 42: please change [Si(OH)4-] to [Si(OH)4] (at seawater pH Si is not ionic), please make this correction elsewhere in the manuscript

- Corrected

Page 4, Line 44: this sentence is basically what Tréguer and De La Rocha (2013) have shown among all oceanic systems. Considering so much dissolution occurs in the water column and sediments (e.g. >95%), the burial of diatoms in sediments is largely predicted by the magnitude of their production in the surface waters. See General Comments.

- Yes, Tréguer and De la Rocha (2013) confirm what had been previously shown by Lisitzin (1971), and our data indicate that the same relation is valid within specific diatom productive environments, such as the coastal upwelling systems.

Page 5, Line 23-24: While this is certainly the conclusion of this analysis, such a statement is rather grandiose given the record is a 47-year average, and later it is discussed that the temporal scale of inference for the silicon-related discussion is tens to hundreds of years (Page 6, line 32). Perhaps be more conservative?

- This comment refers to the sentence "As a whole, the physics (although essential), does not appear as the primary factor determining diatom bloom size and its sediment record."

To calibrate a proxy the sediment material used is always the first top 1 to 2 cm. In coastal regions the sedimentation rate is variable (1.6 to 63 cm/ky), as it can be seen on SI Table 2, what means that the 1 cm of sediment analyzed represents the average conditions of 16 to 500 years. As such, the datasets for any measured parameter that we need to compare our data with should be as long as possible. In the case of the upwelling index we have 47 yr of mean annual values, which is in reality quite good and certainly better than the WOA scarce data existing for nutrients. However, our statement is based on the results obtained with the analysis done for each and all the

different ecological parameters discussed in the paper. Anyway, we have added the temporal resolution to the sentence:

"As a whole and at the scale represented by the sediments (tens – hundreds yr) the physics (although essential), does not appear as the primary factor determining diatom bloom size and its sediment record."

Page 5, Line 37: contrarily vs. contrary

- Corrected

Page 6, Line 7: silicic acid "can be" a limiting nutrient for diatom growth, most kinetic data demonstrate it is unlikely for growth to be limited by silicic acid (also Brzezinski and Nelson 1996 reference was in the North Atlantic gyre, not an upwelling system, see below for upwelling systems references).

- The authors consider that the basic and general references need to be taken into account even if not from the specific region in discussion in this paper.

Page 6, Line 9: the Dugdale and Wilkerson (1998) model was driven by silicic acid nutrient profiles, not direct data, plus this was in the equatorial Pacific not a coastal upwelling system. Additionally, in this same system, Brzezinski et al. (2008, Limnology & Oceanography) showed it was unlikely that silicic acid was limiting diatom growth based on the degree of kinetic limitation observed.

Both environments are characterized by upwelling conditions capable of supporting a productive diatom habitat, and the reference to Dugdale's paper is based on its importance for the understanding of diatoms' physiology and nutrient uptake. Brzezinski et al, 2008 reference was not included because EEP was not the subject of discussion.

Page 6, Line 29: This assumption may be reasonable (see comment on Page 4, Line 44 regarding Tréguer and De La Rocha 2013 study and perhaps cite this as justification) but completely ignores diatom frustules in the sediments which have been authigenically transformed via reverse weathering (e.g. Michalopoulos et al. 2000, Geology 28)

and are likely to not be quantified (i.e. not recognizable). Additionally, the SDA proxy may be more representative of large and/or heavily silicified diatoms only. Perhaps the potential bias could be discussed in the methods. - The potential bias is a reality, and most lightly silicified species are actually dissolved in the water column, as shown by most trap data referred to in the text. However, in all coastal upwelling systems diatom blooms are dominated by species of the genus Chaetoceros, a genus that produces heavily silicified resting spores with high sinking rates and high preservation potential, which leads to a sediment diatom assemblage also dominated by Chaetoceros. A new and illustrative reference to this is now included in the introduction.

The reverse weather is actually discussed within the section 3.1.2

Page 6, Line 34: Dugdale et al. (2011) reference is from equatorial Pacific, not a coastal upwelling system (e.g. see Goering et al. 1973DSR for Peru or Nelson et al. 1981 Consumption and Regeneration of Silicic Acid in Three Coastal Upwelling Systems for Baja California and Northwest Africa). Page 6,Line 37: Goering et al. 1973 actually showed Michaelis-Menten uptake fit Si uptake responses in an upwelling system, the Dugdale paper did not focus on silicate.

- The suggested references are now considered.

Page6, Line 42: expand [S] to be [Si(OH)4]

- Corrected

Page 7, Line 2: use of SisurfMAX is unclear until you mine through supplementary tables

- The authors consider that the explanation should be presented as supplementary information, however we are willing to follow whatever decision the editor considers the best.

Page 7, line 18: I disagree, if iron is the limiting nutrient in the upwelling system (e.g. Bruland et al. 2001, Limnology and Oceanography) then it isn't Silicon uptake which

affects C sequestration, it is iron which leads to excess Si ballast (see also Brzezinski et al. 2015, JGR Oceans).

-Both papers indicated by the reviewer reveal very important results to better understand the importance and role of Fe on the primary production and the algal community composition in two Pacific upwelling systems. However, there is an important aspect that has yet to be considered, the diatom assemblage composition at the different Fe conditions. No clear information is given except that Fe limitation tends to limit the occurrence of large diatoms (> 40 $\mu$m) and the quote in Bruland et al, (2001) "Fe is an important nutrient in limiting the extent of the blooms of large diatoms, such as the very large Coscinodiscus". Coscinodiscus is the genus that contains some of the larger diatom species, 30 – 120$\mu$m in diameter, but those forms are rare in coastal upwelling environments and always a minor component of the sedimentary diatom assemblage. As commented in the introductory text of the manuscript, in coastal upwelling systems the diatom assemblage is dominated by species of the genus Chaetoceros, which are much smaller (3 – 20 $\mu$m), frequently form strains, and produce highly silicified resting spores as a survival strategy, when nutrients became exhausted. Such spores have a high settling rate and are the dominant and best-preserved component of the SDA in all coastal upwelling areas.

Furthermore, although iron limitation is currently considered an important control on phytoplankton growth in the Californian and Peruvian upwelling zones, where there is high delivery rate of macronutrients, the Canary and the Benguela systems are the most silicate depleted of the 5 studied regions (WOA13 show average [Si] to be 1/5 of the Pacific values) although relatively iron-replete (Capote and Hutchins, 2013).

While at global scale Si availability is, as discussed in the discussion paper, determined by inter-ocean fractionation, in the case of Fe there are no inter-oceanic differences. Fe sources (remineralization within the water column, episodic inputs of Fe from continental regions (riverine or eolian input) (Boyd and Ellwood, 2010), re-suspension from the benthic boundary layer and early diagenesis remineralization of sediment Fe and upward diffusion of Fe-enriched pore waters (Johnson et al., 1999); (Masay et al., 2014) are common to all 5 upwelling systems. This fact implies smaller differences for [Fe] in the Atlantic and the Pacific systems as it is demonstrated by the published data (Table 1), even though those data represent a snapshot of [Fe] that varies seasonally and regionally between and within systems.

Furthermore, Sunda and Huntsman (1995) laboratory experiments, indicate that coastal diatoms growth rate is limited for Fe <0.1 nM and becomes optimal at Fe >0.5 nM, while a 0.3 nM Fe threshold for high growth rates of coastal diatoms was found by Bruland et al, (2001). Considering these threshold values and the Fe data compiled in Table A, only the Benguela system values, which were actually measure offshore of the coastal upwelling centers, appears below the threshold for high growth rates of coastal diatoms. That is, on the basis of this analysis, Fe should not be limiting diatom growth rate in coastal upwelling systems.

Anyway, and although the lack of well distributed Fe data prevents the assessment of this micronutrient on SDA, to check the reviewer suggestion that Fe influences SDA via diatom silicification and better preservation in the sediments, we have attempted to examine possible Fe contribution to each coastal upwelling system either by river input or through the interaction of upwelling waters with the bottom Fe-rich fine sediments.

Considering continental Fe input (river or aeolian input), and shallow and broad shelf sediments via resuspension, it is expected that Fe availability increases from open ocean to coastal regions. This is because shelf fine Fe-enriched bottom sediments tend to concentrate in loci of preferential deposition determined by shelf morphology and winter currents direction, but generally in the inner-shelf. Not having the distance to the coast, nor the shelf width in each region, we used water depth at each location as a first approach (Figure 1).

If Fe limitation increased as upwelled waters were advected offshore (Bruland et al, 2001; 2005), and low Fe enhanced Si utilization by diatoms leading to frustule thickness

and better preservation, SDA should decrease with distance from the coast. However, as it can be seen from figure A, SDA shows a large spreading, which is independent of water column depth.

Another way of extrapolating Fe effect could be via freshwater influence / surface waters salinity (Figure 2), but no indication of higher SDA for low salinities is observable. However, the low salinities observed for NE Pacific sites and the southern sites of the SE Pacific (red and rose dots) reveal the important freshwater input by the Columbia River and the Chilean fiords respectively.

Considering that river input is also the main source of Si to the ocean, we have compared the values of salinity and surface water silicate content (Figure 3). A relationship for higher Si at lower salinities in the NE Pacific confirms the importance of the Columbia River as a source of Si and possibly also of Fe, but no relationship to SDA was found (Figure 2).

According to DiTullio et al, 2005, "luxurious Fe uptake near the coast may be important in fueling diatom production as cells are advected off the coast via Ekman transport. Hence, it is likely that these diatoms can remain Fe replete while inhabiting low-Fe waters." If that were the case, then the upwelling of Fe-rich subsurface waters in coastal upwelling systems would be likely to be enough to allow for the development of massive diatom blooms and have a luxurious uptake of Fe up to the point when Fe concentration become below the threshold level and cause a shift of the phytoplankton community. Conversely, (Brzezinski et al, 2015) defend that upwelling brings inadequate iron to the surface for phytoplankton to completely utilize macronutrients, and although the Fe and Si co-limitation of diatom growth, defended by the same author for the HNLC equatorial Pacific system (Brzezinski et al, 2008)(Brzezinski et al., 2008), is lower under strong coastal upwelling, it should increase along aging upwelled waters, that is, along Ekman advection and macronutrients consumption. Furthermore, these authors also defend that low Fe amplifies total net opal production in different ways depending on the Si:N ratio of the initial upwelled waters. That is, higher ballast (diatom species and valves

more silicified and resistant to dissolution) for low Si:N and higher Si production (diatom valves less silicified and resistant to dissolution) for higher Si:N conditions.

Surface waters Si:N at our sites, and its relation to SDA (total number of diatom valves /g of sediment) at all the sites in all 5 coastal upwelling systems is shown in Table 2. No significant relationships were encountered for any of the regions,or all the 5 regions together.

However, if the average and StDev are calculated for SDA and Si:N (from WOA13) for each coastal upwelling system, a highly significant and positive relationship emerges ($R2=0.64$ for a $n=338$ and $p=0.1$). Figure 4 is a plot of those data. Si:N<1 occur off the Galician and Portuguese Margins while for all the other upwelling areas Si:N ratios are >1.

On the basis of Brzezinski et al, (2015) heavily silicified diatoms were to be expected in the Portuguese-Canary System, while in all the other systems, low Fe should enhance total net silica production but of less preservable diatoms. The observation of samples shows dominance of heavily silicified resting spores at all sites, and from the observation of figure D, it becomes clear that the SDA-Si:N relationship throughout the 5 most important coastal upwelling areas follows a trend that is highly similar to the observed between SDA and [Si(OH)4] on surface waters. This leads to the conclusion that independently of [Fe] and its potential effect, total diatom production at any one coastal upwelling system, is determined by the dominant physiological capacity to utilize Si along the number of years that the sediment represents (10 to 100 years).

References

Boyd, P. W. and Ellwood, M. J.: The biogeochemical cycle of iron in the ocean., Nature Geosci, 3, 675:682, 2010.

Bruland, K. W., Rue, E. L., and Smith, G. J.: Iron and macronutrients in California coastal upwelling regimes: Implications for diatom blooms, Limnology and Oceanography, 46, 1661-1674, 2001.

Bruland, K. W., Rue, E. L., Smith, G., and DiTullio, G. R.: Iron, macronutrients and diatom blooms in the Peru upwelling regime: Brown and blue waters of Peru, Mar. Chem., 93, 81-103, 2005.

Brzezinski, M., Krause, J. W., Bundy, R. M., Barbeau, K. A., Franks, P., Goericke, R., Landry, M. R., and Stuke, M. R.: Enhanced silica ballasting from iron stress sustains carbon export in a frontal zone within the California Current, J. Geophys. Res. Oceans, 120, 4654-4669, 2015.

Brzezinski, M., Dumousseaud, C., Krause, J. W., Measures, C., and Nelson, D.: Iron ans silicic acid concentrations together regulate Si uptake in the equatorial Pacific Ocean, Limnol Oceanogr, 53, 875-889, 2008.

Capote, D. G. and Hutchings, D. A.: Microbial biogeochemistry of coastal upwelling regimes in a changing ocean, Nature Geosci, 6, 711-717, 2013.

DiTullio, G. R., Geesey, M. E., Maucher, J. M., Alm, M. B., Riseman, S. F., and Bruland, K. W.: Influence of iron on algal community composition and physiological status in the Peru upwelling system, Limnol Oceanogr, 50, 1887-1907, 2005.

Dugdale, R. and Wilkerson, F. P.: Silicate regulation of new production in the equatorial Pacific Johnson, K. S., Chavez, F. P., and Friederich, G. E.: Continental-shelf sediment as a primary source of iron for coastal phytoplankton, nature, 398, 697-700, 1999.

Hatta, M., Measures, C., Wu, J., Roshan, S., Fitzsimmons, N., and Sedwick, P. N.: An overview of dissolved Fe and Mn Distributions during the 2010-2011 U.S. GEO-TRACES north Atlantic Cruises: GEOTRACES GA03, Deep Sea Research Part Ii, doi: 10.1016/j.dsr2.2014.07.005, 2014. 2014.

Lisitzin, A. P.: Distribution of Siliceous Microfossils in Suspension and in Bottom Sediments. In: The micropaleontology of the Oceans, Cambridge University Press, London, 1971.

Johnson, K. S., Chavez, F. P., and Friederich, G. E.: Continental-shelf sediment as a primary source of iron for coastal phytoplankton, nature, 398, 697-700, 1999.

Masay, C. M., Sedwick, P. N., Dinniman, M. S., Barrett, P. M., Mack, S. L., and McGillicuddy Jr, D. J.: Estimating the benthic efflux of dissolved iron on the Ross Sea continental shelf., Geophys. Res. Lett., 41, 7576:7583, 2014.

Sunda, W. G. and Hunstman, S. A.: Iron uptake and growth limitation in oceanic and coastal phytoplankton, Mar. Chem., 50, 189-206, 1995.

Tréguer, P. J. and De La Rocha, C. L.: The World Ocean Silica Cycle, Annu. Rev. Mar. Sci., 5, 5.1–5.25, doi: 10.1146/annurev-marine-121211-172346, 2013.

[Figure]

**Figure 1 – Relationship between SDA and water depth.**
**Color code as in figure 1 of the discussion version.**

[Figure]

**Figure 2 – Relationship between salinity from the WOA13 data set and SDA. Color code as in figure 1 of the discussion version.**

[Figure]

**Figure 3 – Relationship between salinity and [Si(OH)$_4$]. Data from WOA13
and color code as in figure 1 of the discussion version.**

[Figure]

**Figure 4 - Average and StDev of SDA per upwelling system, considering only the true upwelling areas, *vs* the average and StDev of Si:N estimated from the WOA13 data set. Color code as in figure 1 of the discussion version.**

| Coastal Upwelling System | [Fe] nM | Reference |
|---|---|---|
| Canary System | | |
| Portugal | 0.5 | Hata et al, 2014 |
| Mauritania | 1.5 | Hata et al, 2014 |
| Benguela System | | |
| Distal Antarctic Atlantic sector | <0.3 | Klunder et al, 2011 |
| Humboldt System | | |
| Offshore Peru | < 0.1 | DiTullio et al, 2003 |
| Coastal Central Peru | < 0.4 | DiTullio et al, 2003 |
| Coastal Northern Peru | 0.7 to 1 | DiTullio et al, 2003 |
| California System | | |
| 36 ºN | < 1 | Bruland et al, 2001 |
| N of 37ºN | up to > 10 | Bruland et al, 2001 |
| Somalia-Oman | | |
| | 4 to 7 | Sunda and Huntsman, 1995 |

**Table 1 - [Fe] in the Atlantic and the Pacific coastal upwelling systems from published data**

| Areas | AVERAGE | PEARSON CORRELATION | |
|---|---|---|---|
| | Si:N | n | Si/N |
| Galiza / Portuguese-Canary System | 0.60 | 10 | -0.24 |
| Portuguese Margin /  Portuguese-Canary System | 0.94 | 61 | 0.01 |
| NWAfrica Canary / Portuguese-Canary System | 2.65 | 34 | 0.10 |
| NW Africa / Canary System | 1.33 | 50 | -0.08 |
| SW Africa / Benguela System | 1.58 | 52 | 0.10 |
| SE Pacific / Humboldt System | 1.28 | 162 | -0.05 |
| NE Pacific / California System | 3.24 | 37 | -0.18 |
| Oman / Indian Monsson | 8.75 | 23 | -0.38 |
| TOTAL | 1,90 | 338 | -0.05 |

*p*=.01; p=.05

**Table 2 - Surface waters mean Si:N values at our sites, and its relation to SDA (total number of diatom valves /g of sediment) for all different systems and all the 5 coastal upwelling systems.**

---

## Editor Comment (EC1) · F. Chai (Editor) · 10 Jun 2016

Dear. Dr. Abrantes

I typed a wrong message in my previous email sent to you, somehow due to my internet connection from China. As the Handling Associate Editor for your manuscipt, I apologize to you and your co-authors for delaying my decision of your manuscript.

I have read your replies to both reviewer comments and related changes in the revised manuscript. I am satisfied with your replies and revision of the manuscript. I recommend your final version of the manuscript to be published.

Thank you for contributing to Biogeosicences. I am sorry again for the delay.

Regards, Fei